# Novel Nafion/Graphitic Carbon Nitride Nanosheets Composite Membrane for Steam Electrolysis at 110 °C

**DOI:** 10.3390/membranes13030308

**Published:** 2023-03-07

**Authors:** Taipu Chen, Bo Lv, Shucheng Sun, Jinkai Hao, Zhigang Shao

**Affiliations:** 1Fuel Cell System and Engineering Laboratory, Key Laboratory of Fuel Cells & Hybrid Power Sources, Dalian Institute of Chemical Physics, Chinese Academy of Sciences, Dalian 116023, China; 2University of Chinese Academy of Sciences, Beijing 100039, China

**Keywords:** high temperature proton exchange membrane, steam electrolysis, composite membrane, graphitic carbon nitride, proton conductivity, Nafion, thermal stability, high current density

## Abstract

Hydrogen is expected to have an important role in future energy systems; however, further research is required to ensure the commercial viability of hydrogen generation. Proton exchange membrane steam electrolysis above 100 °C has attracted significant research interest owing to its high electrolytic efficiency and the potential to reduce the use of electrical energy through waste heat utilization. This study developed a novel composite membrane fabricated from graphitic carbon nitride (g-C_3_N_4_) and Nafion and applied it to steam electrolysis with excellent results. g-C_3_N_4_ is uniformly dispersed among the non−homogeneous functionalized particles of the polymer, and it improves the thermostability of the membranes. The amino and imino active sites on the nanosheet surface enhance the proton conductivity. In ultrapure water at 90 °C, the proton conductivity of the Nafion/0.4 wt.% g-C_3_N_4_ membrane is 287.71 mS cm^−1^. Above 100 °C, the modified membranes still exhibit high conductivity, and no sudden decreases in conductivity were observed. The Nafion/g-C_3_N_4_ membranes exhibit excellent performance when utilized as a steam electrolyzer. Compared with that of previous studies, this approach achieves better electrolytic behavior with a relatively low catalyst loading. Steam electrolysis using a Nafion/0.4 wt.% g-C_3_N_4_ membranes achieves a current density of 2260 mA cm^−2^ at 2 V, which is approximately 69% higher than the current density achieved using pure Nafion membranes under the same conditions.

## 1. Introduction

Hydrogen energy is expected to be of significant importance over the next 30 years, and the global hydrogen industry is projected to be worth 10 trillion dollars by 2050 [1]. Wind and solar power are distributed energy sources, which are difficult to deploy on the grid. This is due to the intermittent and unstable nature of renewable energy, which leads to a huge gap between energy supply and consumption [2]. However, they can be used to produce hydrogen, which is an efficient energy carrier, via economical and environmentally friendly methods. Hydrogen can be produced via water electrolysis, which is nonpolluting, does not require separation operations, and can be adjusted to suit the required hydrogen−production capacity [3].

Recent studies on water electrolysis have concentrated on proton exchange membrane (PEM) electrolysis for alkaline electrolytic cells, which produce high−purity hydrogen, operate at high electrical densities, exhibit transient responses, and utilize neutral water feeds [4,5,6,7,8,9,10]. However, the performance of PEM electrolysis must be improved to facilitate widespread commercial applications. To exploit the kinetics under high−temperature conditions, PEM electrolysis should be conducted at temperatures higher than 100 °C [11,12,13,14,15,16,17]. As the temperature increases, the ohmic losses of high−temperature PEM water electrolyzers (HT−PEMWEs) and the activated losses of the electrodes are reduced. Therefore, high working temperatures may significantly reduce the overpotential of the electrolyzers and improve their electrical properties. Furthermore, as the temperature increases, the total thermodynamic energy required for water decomposition decreases, which reduces the amount of the electrical energy required. Therefore, at high temperatures, the thermal energy of the operating environment is primarily used to fulfill the activation energy requirement; therefore, this technology can utilize waste heat from industry.

Another advantage of HT−PEMWEs is that no liquid water is used in the electrolyzer, which optimizes the management of water and facilitates subsequent gas compression [18]. However, high temperatures increase the saturation pressure of water and reduce the humidity within the electrolytic cell. Therefore, PEMs may become dehydrated when operated at high temperatures, which will reduce the proton conductivity and degrade the electrochemical properties [19].

Researchers [15,18,20,21,22] have developed novel modified membranes using Nafion to address the issue of a reduced proton conductivity in membranes. There are two main improvement strategies: one is to dope inorganic or porous materials to improve the water absorption of the membrane. The other is to choose other perfluoroalkylsulfonic acid (PFSA) type membranes. Aquivion, for example, has a high softening temperature due to its short sulfonic acid side chain structure. Hansen et al. [15] applied a Nafion membrane doped with phosphoric acid in a HT−PEMWEs. In comparison to conventional perfluorosulfonic acid membranes, their membrane exhibited good thermostability and cell performance under high−temperature conditions. However, issues such as severe phosphoric acid loss and toxic effects on the catalyst remain to be resolved. Aili et al. [18] developed Nafion−blended membranes using polybenzimidazole (PBI). The phosphate absorption rate of the membranes was mainly affected by the ratio of PBI to Nafion, and it was minimized using a neutral blend. Antonucci et al. [20] employed SiO_2_−doped modified−Nafion membranes for steam electrolysis applications. The modified membranes performed significantly better than commercial Nafion membranes, owing to their superior water uptake abilities to that of pristine membranes. Baglio et al. [21] reported that TiO_2_−doped modified−Nafion membranes showed promising results in electrolytic cells operating at high temperatures. They performed better than Nafion 115, primarily because of the water−preservation properties of TiO_2_. There are also researchers studying other polymers to make HT−membranes. Diaz−Abad et al. [23] developed a polybenzimidazole/graphene oxide (PBI/GO) electrolyte for hybrid sulfur (HyS) cycling with a theoretical voltage of 0.16 V. Yin et al. [24] prepared phosphoric acid−doped PVC−P4VP membrane by cross−linking polyvinyl chloride (PVC) and poly(4vinylpyridine) (P4VP). Kamaroddin et al. [25,26] experimented with PBI/ZrP (zirconium phosphate) as a membrane for PEM electrolyzers. It showed higher efficiency of hydrogen production compared to Nafion 117. Park et al. [27] developed partially sulfonated poly(arylene ether sulfone) (sPAES) for the membrane. At 50% sulfonation, the electrolyte of the same thickness exhibited lower hydrogen permeation than Nafion.

Graphitic carbon nitride (g-C_3_N_4_) has the same layer−like construction as graphene material, and it is widely used in catalysts owing to its excellent thermal and hydrothermal (hydrothermal) stability and simple synthesis [28,29,30]. The g-C_3_N_4_ surface contains basic sites owing to the −amino and −imino groups. These basic sites can interact with the −SO_3_H group, which enhances the proton conductivity [31,32,33,34]. Several studies have investigated the properties of sulfonated poly(ether ether ketone) (SPEEK)/g-C_3_N_4_ composite membranes in vanadium oxide liquid flow batteries (VRFBs) [35,36], and the composite membranes exhibited high selectivity and low vanadium permeability. Furthermore, g-C_3_N_4_/SPEEK composite membranes have been applied to direct methanol fuel cells (DMFCs), where they improved the power density compared to that of conventional DMFCs [37]. However, the application of g-C_3_N_4_ in proton exchange membrane steam electrolysis has not been investigated.

However, no studies have applied g-C_3_N_4_ to PEM steam electrolysis. Therefore, this study aimed to prepare Nafion/g-C_3_N_4_ modified membranes by solution casting with g-C_3_N_4_ as the modified filler. The structure and physical properties of the membranes were characterized, and the mechanisms underlying the enhanced proton conductivity were investigated.

## 2. Materials and Methods

### 2.1. Materials

Sulfuric acid (95–98%, analytical reagent (AR)) was purchased from Bonuo (Dalian, China). Dicyandiamide was purchased from Macklin. Isopropanol (AR) and dimethyl sulfoxide (DMSO) were purchased from Kermel (Dalian, China). Nafion D520 ionomer was purchased from DuPont (USA).

### 2.2. Preparation of g-C_3_N_4_ Nanosheets

g-C_3_N_4_ was produced as shown in Figure 1. Dicyandiamide (5 g) was heated in a tube furnace from room temperature at a heating rate of 5 °C min^−1^ and calcined at 550 °C for 4 h under an air condition. After cooling naturally, a yellow solid was obtained. Thereafter, g-C_3_N_4_ was produced by pulverizing the resulting product into a powder [38].

### 2.3. Fabrication of the Nafion/g-C_3_N_4_ Membrane

The Nafion/g-C_3_N_4_ composite membranes were produced using the solvent−molding technique, as shown in Figure 1. Firstly, 100 g of Nafion D520 ionomer was vaporized at 80 °C. Thereafter, 95 g of DMSO was used to dissolve the resin to obtain a 5 wt.% Nafion solution. g-C_3_N_4_ was dispersed in 16 g of the Nafion solution and subjected to vigorous sonication. The g-C_3_N_4_ mass fraction (relative to Nafion) in the composite membranes varied from 0–1.0%. The Nafion/g-C_3_N_4_ mixtures were inverted onto recess plates and dried in an oven at 80 °C for 24 h and at 120 °C for 12 h. The samples were naturally cooled to 25 °C, and then, the modified membranes were obtained by peeling them from the plates. The membranes were immersed in 0.5 mol L^−1^ H_2_SO_4_ at 80 °C for 3 h, and then heated in ultrapure water at 80 °C for 1 h, which was repeated for three times. Finally, the membranes were air−dried for further study. The membranes were labeled Nafion/X wt.% g-C_3_N_4_ depending on the mass of g-C_3_N_4_ doping. The thicknesses of the composite membranes were 49−53 μm, which was similar to that of the Nafion 212 membranes (50.8 μm).

### 2.4. Physical Characterization

The micromorphologies of the g-C_3_N_4_ nanosheets and modified membranes were observed using transmission electron microscopy (TEM, Tecnai G2 F30) and scanning electron microscopy (SEM, SU8020). The crystal structures of g-C_3_N_4_ and the modified membranes was examined using X−ray diffraction (XRD, Empyrean−100). The chemical moieties of the g-C_3_N_4_ and modified membranes were examined using Fourier−transform infrared (FTIR) spectroscopy (Nicolet iS50). The thermal stability was examined via the thermogravimetric analysis (TGA, STA449F5) with a warming rate of 10 °C min^−1^ under a N_2_ atmosphere. The mechanical strengths of the modified membranes were examined using a universal testing machine (WDW−10).

### 2.5. Water Absorption and Degree of Swelling

The amount of water absorbed by a membrane can be determined by comparing its wet and dry masses. To determine the wet mass of the membrane, the membrane was first soaked in ultrapure water at 30 °C for 24 h; thereafter, the surface of the membrane was dried rapidly using filter paper before the sample was weighed. To determine the dry mass of the membrane, the membrane was dried in an oven at 80 °C for 24 h. The water absorption rate was calculated using the Equation (1):(1)$$\mathrm{Water}\;\mathrm{absorption}\;\left(\%\right)=\frac{M_{\mathrm{wet}}-M_{\mathrm{dry}}}{M_{\mathrm{dry}}}\times 100\%$$
where M_wet_ and M_dry_ are the wet and dry masses of the membrane, respectively.

To determine the degree of swelling, the dry membranes were immersed in ultrapure water for 24 h. The degree of swelling was calculated by comparing the dry and wet areas of the membranes based on the Equation (2):(2)$$\mathrm{Degree}\;\mathrm{of}\;\mathrm{swelling}\;\left(\%\right)=\frac{A_{\mathrm{wet}}-A_{\mathrm{dry}}}{A_{\mathrm{dry}}}\times 100\%$$
where A_wet_ and A_dry_ are the wet and dry membrane areas, respectively.

### 2.6. Proton Conductivity Tests

The proton conductivities of the membranes were determined via electrochemical impedance spectroscopy (EIS) with a Gamry Interface 5000E with a frequency range of 1 MHz to 1 Hz and at a measurement potential amplitude of 10 mV.

For working temperatures of 30–90 °C, the membranes were maintained at the working temperature for 20 min before the proton conductivity tests (to reduce measurement errors), and then, they were soaked in ultrapure water. The proton conductivity (*σ*) of these samples were determined using the Equation (3):(3)$$\sigma =\frac{L}{RWd}$$
where *L* is the distance between the two electrodes; *R* is the resistance of the sample; *W* is the width of the sample; and *d* is the single membrane thickness.

For working temperatures of 100–120 °C, the proton conductivity in the direction of membrane penetration was tested. The uncoated catalyst membrane was sandwiched between two gas diffusion layers (GDLs) and made into a membrane electrode assembly by hot pressing. The effective area of the sample was 5 cm^2^. Tests were conducted at ambient pressure using the HT−PEMWE setup with a water intake of 15 cm^3^ min^−1^. The proton conductivities of these samples were determined using the Equation (4):(4)$$\sigma =\frac{L}{RA}$$
where *L* is the thickness of a single membrane; *R* is the resistance of the sample; and *A* is the area of the electrode.

### 2.7. Electrolysis Tests

The high−temperature PEM steam electrolysis polarization curves were recorded using a custom−built electrolysis platform, as shown in Figure 2a. Steam was supplied to the anode from an evaporator at 180 °C, and the inlet air path was insulated with heating tape at 140 °C. Water was pumped into the evaporator at a rate of 15 cm^3^ min^−1^, and it evaporated completely. The electrolyzer was warmed by heating bars to the working temperature for 30 min before steam was introduced into the electrolyzer. The electrolysis tests were conducted at 110 °C and ambient pressure. The effective area of the electrolyzer is 5 cm^2^. Ir black was used as the anode catalyst, 70 wt.% Pt/C was used as the cathode catalyst, and Nafion was used as the binder. The Ir loading was 1.5 mg cm^−2^ and the Pt loading was 0.5 mg cm^−2^. The anode gas diffusion layer was Pt−plated Ti felt, and carbon paper was used as the cathode gas diffusion layer. Gas diffusion electrodes (GDEs) are prepared by covering the catalyst with GDLs using a spraying method. The PEM is placed between two GDEs and held in place by polyester frames, which is thermally pressed to form a membrane electrode assembly (MEA). The flow field of the bipolar plates is a parallel flow field. The cell is assembled in the structure shown in Figure 2b.

## 3. Results

### 3.1. Characterization of g-C_3_N_4_

g-C_3_N_4_ has a clear two−dimensional sheet−like morphology with minimal aggregation (Figure 3a–c). Energy−dispersive spectroscopy (EDS) analysis indicates that N is uniformly distributed throughout the g-C_3_N_4_ nanosheets (Figure 3d). The elemental composition of g-C_3_N_4_ is determined using EDS. g-C_3_N_4_ is composed mainly of C (42.98 at.%) and N (57.02 at.%) with a C/N ratio of 0.754 (approximately equal to the theoretical figure of 0.75). These results indicate the effective synthesis of g-C_3_N_4_.

The crystal structure of g-C_3_N_4_ is shown in Figure 4a. Two strong absorption peaks are observed in the g-C_3_N_4_ spectrum at 27.43° and 12.78°, which corresponded to (002) and (100) planes, respectively. These results are in agreement with those reported by Yuan et al. [39], and they indicate that g-C_3_N_4_ was successfully synthesized [35,36].

Figure 4b shows the FT−IR spectrum of g-C_3_N_4_, confirming the production of g-C_3_N_4_. The strong peak at 803 cm^−1^ is the signature peak of the triazine ring, and the characteristic peaks between of 1250 and 1650 cm^−1^ correspond to the tensile peaks of N−(C)_3_ and C−NH−C [40,41]. The broad peak at approximately 3156 cm^−1^ is associated with the defective positions of the primary and secondary amines and their intermolecular hydrogen bonding interactions [42].

### 3.2. Properties of Nafion/X wt.% g-C_3_N_4_ Modified Membranes

The dispersion of g-C_3_N_4_ in Nafion is investigated using SEM, and elemental distribution is detected using EDS. g-C_3_N_4_ are evenly dispersed with negligible aggregation in Nafion, as shown in Figure 5, owing to the good surface interactions between g-C_3_N_4_ and Nafion and the large surface area of g-C_3_N_4_ [43]. Furthermore, the uniform g-C_3_N_4_ distribution enhances the tensile strength of the membrane, which was consistent with the test results for the modified membranes. Doping ratios higher than 1.0% have also been experimented. However, excessively high doping ratios can cause significant agglomeration of g-C_3_N_4_, which severely decreases the mechanical properties and homogeneity of the membrane, and even prevents the formation of the membrane.

Two peaks at 2θ ≈ 17° and 2θ ≈ 39° were present in all the modified membranes (Figure 6a). The broad peak at 2θ ≈ 17° is composed of two peaks and can be decomposed into two peaks at 2θ = 16.6° and 2θ = 17.3°. Information about the morphology of the main chain in Nafion is represented by a peak at 2θ ≈ 16° and the crystallinity of the main chain is represented by a peak at 2θ ≈ 17.5° [44]. Therefore, the XRD patterns show that the addition of g-C_3_N_4_ had negligible effects on the crystallinity of the Nafion membranes.

Figure 6b shows the FT−IR spectra of the modified membranes. For the Nafion (perfluorosulfonic acid) membranes, the typical absorption vibration peaks originate from the −SO_3_ group and the C−O−C groups. The stretching vibration absorption peak of −CF_2_−CF_2_ appeares at 1150 cm^−1^ and the typical peaks of −SO_3_ and C−O−C on the side chains are observed at 1056 and 982 cm^−1^ [45,46]. After doping with g-C_3_N_4_, the intensity of the peak between 1130 and 1250 cm^−1^ reduced, which may be caused by C−F···H−N hydrogen bonding.

As shown in Figure 7, doping with g-C_3_N_4_ enhances the thermostability of the modified membrane. All the membranes exhibit a typical weight−reducing pattern [47,48]. The weight loss below 140 °C occurred as residual water was lost from the membranes. Between 140–250 °C, no significant weight loss was noted, and the membranes appeared to be stable. For pure Nafion membranes, the first loss at 250–350 °C occurred owing to the disconnection of the sulfonic acid group, whereas the second loss at 380–450 °C occurred owing to side−chain degradation. By contrast, for the composite membranes, the first loss interval did not begin until 340 °C. The mass loss at 340–400 °C is mainly caused by the decomposition of the side chains. As shown in Figure 7a, the main loss interval (accounting for approximately 70% of the total loss) in all the membranes occurs above 450 °C. This loss was caused by the thermal degradation of the Teflon backbone. Derivative thermogravimetric (DTG) analysis is shown in Figure 7b. In summary, the acid–base interaction between g-C_3_N_4_ and Nafion improves the thermostability of the modified membranes compared to that of the pure Nafion membrane [49].

### 3.3. Water Absorption, Swelling Degree, and Mechanical Properties

Appropriate water absorption can enhance the proton conductivity of the membranes and facilitate electrolysis at high temperatures. As presented in Table 1, g-C_3_N_4_ doping significantly increases the amount of water absorbed by the membranes, which occurs because of the strong attraction between g-C_3_N_4_ and the water molecules. The water absorption rate of the composite membrane with 0.8 wt.% doping is 30.65%, which is approximately 74% higher than that of pure Nafion. The increased g-C_3_N_4_ content is expected to form larger ion clusters and produce more hydrogen bonds, which can enhance proton transport.

The mechanical properties of the membranes are listed in Table 1. The addition of g-C_3_N_4_ increases the mechanical stability. As shown in Figure 8, the elongation at the breaking point of the pure Nafion membrane is 135.15%, and the tensile strength is 12.03 MPa. By comparison, the modified membranes had a higher tensile strength and lower elongation at the breaking point. The Nafion/0.8% g-C_3_N_4_ membrane has the highest tensile strength of 14.63 MPa, which is approximately 22% higher than that of pure Nafion. The test results indicate that g-C_3_N_4_ doping can be used to enhance the mechanical properties of the membrane.

### 3.4. Proton Conductivity

Proton conductivity is an important characteristic that determines the electrolytic properties of the membranes; therefore, it is evaluated using EIS. The proton conductivities of the Nafion/g-C_3_N_4_ modified membranes are higher than those of pure Nafion (Figure 9a). In particular, the Nafion/0.4% g-C_3_N_4_ membrane has the highest proton conductivity of 287.71 mS cm^−1^ at 90 °C, which is approximately 89% higher than that of the pure Nafion. This may be because the modified membranes absorbed more water, which formed proton transport channels. However, the proton conductivity of the Nafion/0.8% g-C_3_N_4_ membrane decreased, probably because the excess g-C_3_N_4_ blocked some of the proton transport channels. Above 100 °C, the relative humidity is low, even when the water vapor feed was at its highest, which reduced the conductivity of the membrane. However, the modified membranes exhibited higher conductivity and did not exhibit abrupt decreases in conductivity at high temperatures (Figure 9b). This indicates that the strong water absorption of g-C_3_N_4_ effectively improves the membrane performance under low humidity conditions. The tests above 100 °C are performed at ambient pressure. Therefore, further improvements in the device to allow testing under pressurized conditions may reveal a significant increase in performance.

### 3.5. Electrolysis Performance

The steam electrolysis performances of the Nafion/g-C_3_N_4_ modified membranes were measured, and the results are shown in Figure 10. At low current densities, the electrolysis voltages of the different membranes were approximately the same, suggesting that they had the same catalytic properties and that the differences in their electrolytic performances were caused by their different conductivities. The commercial Nafion 212 membrane shows approximately the same results as the pure Nafion membrane that we produced. This indicates that the improvement in the electrolytic performance is derived from the addition of g-C_3_N_4_. At 2.0 V, the optimal peak current density of the Nafion/0.4% g-C_3_N_4_ and pure−Nafion membranes are 2260 and 1340 mA cm^−2^, respectively. At a current density of 1000 mA cm^−2^, the Nafion/0.4 %g-C_3_N_4_ membrane has the lowest electrolysis voltage of 1.684 V. The results suggest that doping with g-C_3_N_4_ facilitates electrolysis, which is attributed to the higher proton conductivity of the modified membranes than that of the pure−Nafion membranes. Table 2 lists the high−temperature PEM electrolysis performance of the Nafion/0.4% g-C_3_N_4_ membrane produced in this study compared to the results obtained in other studies. The Nafion/g-C_3_N_4_ membrane exhibits better electrolytic behavior with a relatively low catalyst loading, which demonstrates the superiority of Nafion/g-C_3_N_4_ modified membranes.

## 4. Conclusions

In this study, a strategy for preparing Nafion−composite membranes using g-C_3_N_4_ as a filler is developed. The prepared membranes achieve excellent results when utilized as steam electrolytic PEMs. Observations of the microscopic morphology reveal that the g-C_3_N_4_ is uniformly distributed in the Nafion, which resulted in a high proton conductivity and good mechanical properties. TGA reveal that the addition of g-C_3_N_4_ enhances the thermostability of the membranes. At 90 °C, the highest proton conductivity of the Nafion/0.4% g-C_3_N_4_ membrane is 287.71 mS cm^−1^ compared to 151.86 mS cm^−1^ for pure Nafion. At temperatures higher than 100 °C, the modified membranes exhibit high conductivity, and no abrupt decreases in conductivity are observed. Compared to the materials developed in previous studies, the membranes in this study demonstrated a better electrolytic performance with relatively low catalyst loading. At 2.0 V, the optimum peak current density of the Nafion/0.4% g-C_3_N_4_ membrane is 2260 mA cm^−2^, which is approximately 69% higher than that of pure Nafion (1340 mA cm^−2^). Overall, these results indicate that Nafion/g-C_3_N_4_ membranes have considerable potential for practical applications in PEM steam electrolysis.

## Figures and Tables

**Figure 1 membranes-13-00308-f001:**
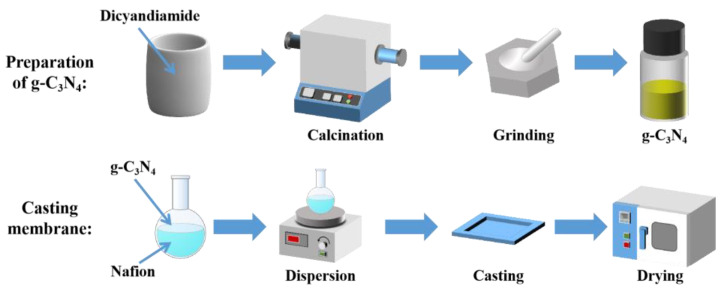
Schematic of the material preparation process.

**Figure 2 membranes-13-00308-f002:**
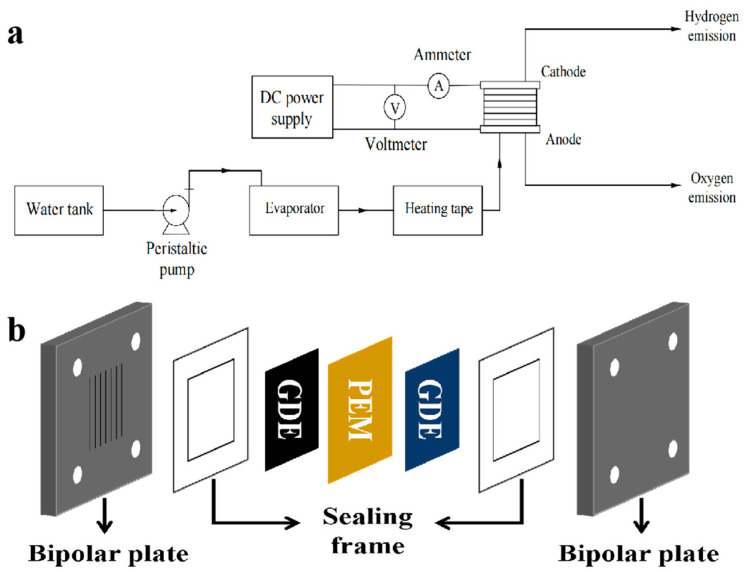
Schematic of (**a**) the HT−PEMWE setup and (**b**) cell structure.

**Figure 3 membranes-13-00308-f003:**
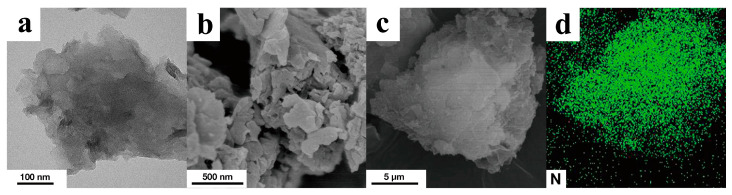
(**a**) TEM image, (**b**,**c**) SEM images, and (**d**) N elemental distribution in the same region of a g-C_3_N_4_ sample.

**Figure 4 membranes-13-00308-f004:**
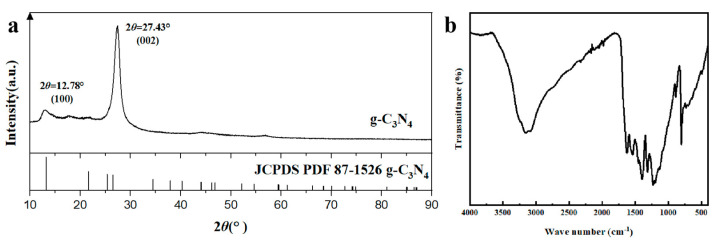
(**a**) XRD and (**b**) FT−IR profiles of the g-C_3_N_4_ nanosheets.

**Figure 5 membranes-13-00308-f005:**
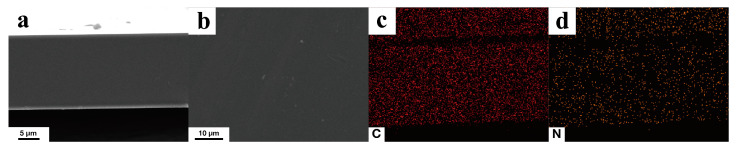
(**a**) Cross−sectional morphology, (**b**) surface morphology, (**c**) C distribution, and (**d**) N distribution in the same region of the Nafion/1.0 wt.% g-C_3_N_4_ modified membrane.

**Figure 6 membranes-13-00308-f006:**
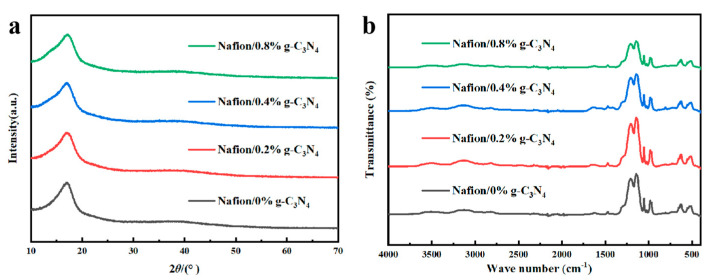
(**a**) XRD and (**b**) FT−IR profiles of the Nafion/g-C_3_N_4_ modified membranes.

**Figure 7 membranes-13-00308-f007:**
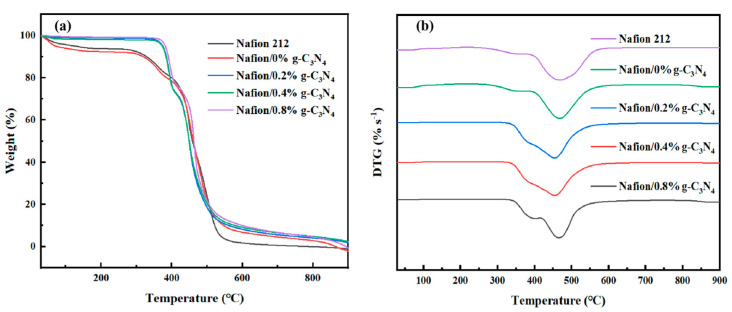
(**a**) TGA and (**b**) DTG diagrams of Nafion/g-C_3_N_4_ modified membranes.

**Figure 8 membranes-13-00308-f008:**
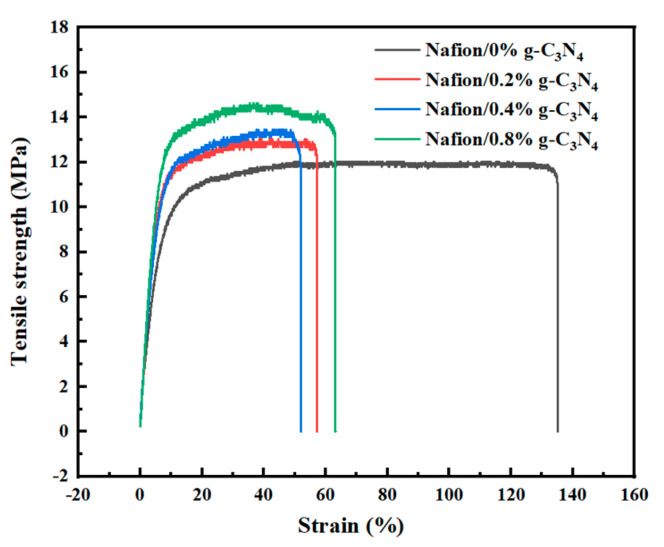
Stress−strain curves for the Nafion/g-C_3_N_4_ modified membranes.

**Figure 9 membranes-13-00308-f009:**
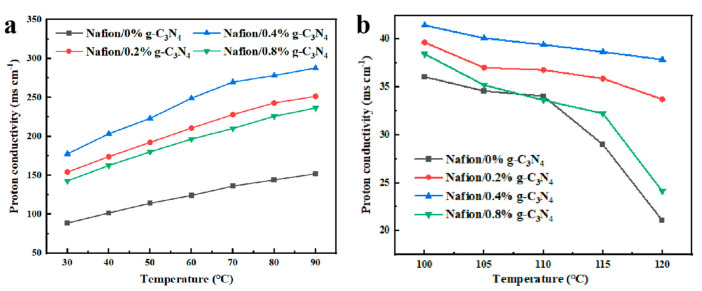
Proton conductivity of the Nafion/g-C_3_N_4_ modified membranes at (**a**) 30–90 °C and (**b**) 100–120 °C.

**Figure 10 membranes-13-00308-f010:**
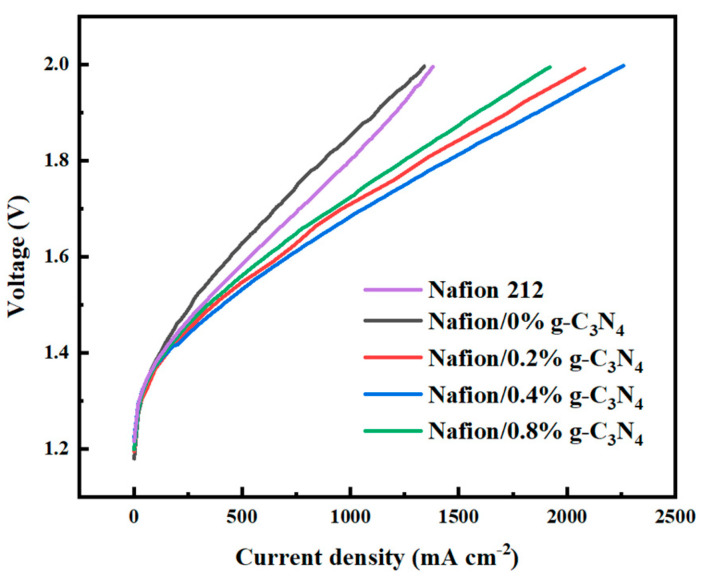
Steam electrolysis performance of the Nafion/g-C_3_N_4_ modified membranes at 110 °C.

**Table 1 membranes-13-00308-t001:** Water absorption, degree of swelling, and mechanical properties of the Nafion/ g-C_3_N_4_ modified membranes.

Membranes	Water Absorption (%)	Degree of Swelling (%)	Tensile Strength (MPa)	Elongation at Break (%)
Nafion/0% g-C_3_N_4_	17.61	15.50	12.03	135.15
Nafion/0.2% g-C_3_N_4_	21.88	16.88	13.02	57.20
Nafion/0.4% g-C_3_N_4_	26.87	18.25	13.44	52.06
Nafion/0.8% g-C_3_N_4_	30.65	23.63	14.63	63.20

**Table 2 membranes-13-00308-t002:** Comparison of the steam electrolysis performance of various membrane materials.

Membranes	Peak Current Density (A cm^−2^)	Anode Catalyst Quantity (mg cm^−2^)	Operating Temperature (°C)	Reference
Nafion/0.4% g-C_3_N_4_	2.26 @2 V	1.5	110	This study
Nafion 212	1.38 @2 V	1.5	110	This study
Nafion/PA	0.3 @1.75 V	4	130	[18]
PBI/PA	0.5 @1.75 V	4	130	[18]
Nafion−SiO_2_	1.18 @1.9 V	5	110	[20]
Nafion−TiO_2_	0.7 @1.8 V	5	120	[21]
Aquivion/PA	0.775 @1.8 V	1	130	[50]

## Data Availability

Data available on request due to restrictions. The data presented in this study are available on request from the corresponding author.

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
