# Peer review of "Novel Nafion/Graphitic Carbon Nitride Nanosheets Composite Membrane for Steam Electrolysis at 110 °C"

_membranes, 2023, doi:10.3390/membranes13030308_

Round 1
Reviewer 1 Report
Water electrolysis is an efficient and green way to produce hydrogen. The key factor of PEM water electrolysis membrane material.
In this paper, C3N4 is combined with Nafion, and good experimental results are obtained.At 2.0 V, the optimum peak current density of the Nafion/0.4 % g-C3N4 301 composite membrane is 2260 mA cm-2 , which is nearly 69 % higher than the 1340 mA cm- 302 2 of the pure Nafion.
My review comments are as follows:
Can author show the sample preparation process in diagram form?
More detailed test system should be added.
Reviewer 2 Report
In this manuscript, A novel composite membrane, fabricated from graphitic carbon nitride (g-C3N4) and Nafion, is firstly used in steam electrolysis and achieves excellent performance. After carefully reading, I would like to recommend this paper be accepted with minor revision. Specific recommendations are listed below.
1. in the introduction, more latest references had better be added.
2. The scale in Figure 2b is incorrect. Meanwhile, two-dimensional sheet of g-C3N4 is not clearly.
3. The scale in Figure 4a is not correct.
4. Please confirm all the used font of units in the text, such as "℃", etc.
5. The paper has some grammatical and spelling errors, which need to be improved and corrected. For example, "g-C3N4" on page 2, line 83, should be subscripted, etc.
Reviewer 3 Report
The manuscript reported modified composite Nafion membranes for PEM steam electrolysis operations above 100 ℃, which are fabricated from graphitic carbon nitride and Nafion. The g-C3N4 doping improves the thermal stability of the modified membranes and also enhances proton conductivity. Above 100 ℃, the modified membranes still exhibit high conductivity and also avoid the sudden drop in conductivity at higher temperatures. It can be concluded that the Nafion/g-C3N4 composite membranes exhibit excellent behaviour in the steam electrolyzers.
I consider the content of this manuscript meets the reading interests of the readers of the Membranes journal. However, there are certain English spelling and grammar issues, and also the discussion and explanation should be further improved. I suggest giving a minor revision and the authors need to clarify some issues or supply some more experimental data to enrich the content.
1. For grammar issues, it is suggested that the author double-check the small grammar errors in the full text, especially the lack of and redundant use of definite articles.
2. For the Keywords, ‘Nafion’, ‘thermal stability’, and ‘high current density’ should be added in order to attract a broader readership.
3. Page 1, ‘Wind and solar power generation have the problem of distributed electricity which is difficult to deploy into the grid.’ What is the root cause of the difficulty in grid connection? There seems to be an insufficient explanation here. It should be noted that the unique intermittence and instability of renewable energy have brought major challenges to the stable operation of the power system, opening temporal and spatial gaps between the consumption of the energy by end-users and its availability (Batteries, 2022, 8(11): 202).
4. Page 2, ‘In addition, a critical advantage of HT-PEMWE is the absence of liquid water in the electrolyzer, greatly optimizing water and gas management and facilitating system and subsequent gas compression [16].’ If there is no liquid water, it is correct that water management can be simple. However, what optimization can this bring to gas management? This issue should be further clarified since it makes confusion to the readers. I consider gas optimization may be more complicated.
5. Page 2, ‘It has also been suggested that proton exchange membranes can become dehy-drated during high-temperature operation, leading to a reduction in proton conductivity and electrochemical properties [17]. The high temperature increases the saturation pres-sure of the water leading to a reduction in the humidity within electrolytic cells, leading to a reduction in the water absorption and proton conductivity.’ These two sentences should be merged since the reduction of proton conductivity appears twice, and the reduction in the humidity and dehydrated is indeed the same phenomenon.
6. Page 3, ‘DCCY (5 g) was calcined in a tube furnace at a warming speed of 5 ℃ min-1 for 4 h under air conditions.’ I consider the starting and ending temperatures should be given. The melting point of DCCY is 209.5 °C, and 4h means 240 min, the temperature should reach more than 1000 °C. Will the degradation of DCCY take place? And which reaction will happen at so high temperatures?
7. Page 3, ‘The g-C3N4 mass fraction (relative to Nafion) in the composite membranes varies from 0-1.0 %.’ Why is the maximum mass fraction selected as 1.0%? What will happen if 5% or 10% is used?
8. Page 4, ‘Figure 1. Schematic overview of the HT-PEMWE setup.’ Figure 1 should appear after the paragraph which describes the contents of Figure 1. The same applies to Figure 2, which should not appear at the very beginning after ‘3.1 Section’. Double-check Figures 4, 5, 6, 7, 8, and 9.
9. Page 6, ‘The weight change before 140 ℃ is due to the loss of residual water in the membranes, after which membranes appear to be stable with no significant weight loss. ’ No, the stable trend finishes when 500 ℃ arrives, and the ending temperature for the stable range should be given as well. Also for the TGA test, the thermal events are hard to read from the results. Hence, I suggest adding the first derivative of the wt% ascribed to mass loss as well, see Figure 1b and 1c of (Electrochimica Acta, 2019, 309: 311-325).
10. Page 8, ‘Table 2. This is a table. Tables should be placed in the main text near the first time they are cited.’ The real description of Table 2 is missing, and the contents in the current version should be totally changed. Moreover, the results of commercial or pristine Nafion should also be listed in this table.
